# Design and Analysis of an Aerostatic Pad Controlled by a Diaphragm Valve

**Federico Colombo** , **Luigi Lentini \*** , **Terenziano Raparelli** , **Andrea Trivella** and **Vladimir Viktorov**

Politecnico di Torino, Corso Duca degli Abruzzi 24, 10129 Turin, Italy; federico.colombo@polito.it (F.C.); terenziano.raparelli@polito.it (T.R.); andrea.trivella@polito.it (A.T.); vladimir.viktorov@polito.it (V.V.)
\* Correspondence: luigi.lentini@polito.it

**Abstract:** Because of their distinctive characteristics, aerostatic bearings are particularly suitable for high-precision applications. However, because of the compressibility of the lubricant, this kind of bearing is characterized by low relative stiffness and poor damping. Compensation methods represent a valuable solution to these limitations. This paper presents a design procedure for passively compensated bearings controlled by diaphragm valves. Given a desired air gap height at which the system should work, the procedure makes it possible to maximize the stiffness of the bearing around this value. The designed bearings exhibit a quasi-static infinite stiffness for load variation ranging from 20% to almost 50% of the maximum load capacity of the bearing. Moreover, the influence of different parameters on the performance of the compensated pad is evaluated through a sensitivity analysis.

**Keywords:** aerostatic bearings; compensation; infinite stiffness; diaphragm valve; design

## 1. Introduction

Because of their distinctive characteristics, aerostatic bearings are particularly suitable for high-precision applications. Their zero friction, cleanness and long life make aerostatic bearings successful in applications such as machine tools, measuring machines and power board testing [1]. In fact, this type of bearing significantly increases the measuring accuracy of this type of measuring system since it does not generate additional vibrations with respect to the more conventional ball bearings [2,3].

However, because of the compressibility of the lubricant, this kind of bearing is characterized by low relative stiffness and poor damping. For these reasons, many solutions have been proposed since the 1960s, which was a pivotal period in the history of gas lubrication [4].

The first attempts that were made to increase aerostatic bearing stiffness consisted of many sensitivity analyses concerning the type, location and size of the supply hole [5]. Additionally, this type of investigation was further enhanced with the aid of more accurate numerical models. Boffey et al. [6] and Colombo et al. [7] performed numerical and experimental studies to assess the effect of the supply pressure and orifice size and location on the performance of rectangular aerostatic pads. In these works, it was found that choosing a lower nominal air gap height led to slightly higher stiffness.

The same result was found by the adoption of porous or grooved surfaces. In fact, the higher performance that stems from the use of these kinds of feeding systems can be exploited in the presence of lower nominal air gap heights. Moreover, porous and grooved bearings present further relevant drawbacks. Groove geometries have to be suitably selected to avoid pneumatic hammer [8–10], whereas porous surfaces have some critical issues related to the choice of material, which should have both suitable porosity and impact toughness [11,12].

Subsequently, technological improvements in the field of manufacturing and electronics have made it possible to enhance aerostatic bearing performance by means of

compensation methods. These methods consist of the use of additional devices and/or components that, when integrated with aerostatic pads, can enhance performance. Although there are many compensation strategies, e.g., modifying the feeding system of the pad or its geometry, the main goal is to compensate for air gap variations. Compensation methods can be active or passive. In passive compensation methods, bearings are integrated with devices and/or components that exploit only the energy associated with the supply pressure, e.g., pneumatic valves and compliant elements. Conversely, actively compensated bearings are integrated with elements such as sensors, controllers and actuators that require external sources of energy to function [13]. Looking at the current solutions, in actively compensated bearings, the presence of closed-loop systems makes it possible to obtain significant improvements in both static and dynamic performance. Because of their high dynamics, ease of integration and high power density, piezoelectric actuators are used in many active compensation systems. Al-Bender [14] and Aguirre [15] proposed an active compensation solution that makes it possible to exploit different types of conicity control. Here, a circular and centrally fed aerostatic pad is integrated with a (Proportional Integrative) PI controller, displacement capacitive sensors and three symmetrically and circumferentially placed piezo actuators. The integration of these elements with the pad makes it possible to suitably modify the conicity of the air gap in order to compensate for air gap variations. Similarly, Maamari et al. [16,17] proposed another type of conicity control by using a magnetic actuation and internal servomechanism. Adopting a geometrical compensation (also called a support control [14]) is another option that has been proposed by Colombo et al. [18] and Matsumoto et al. [19,20]. In this compensation strategy, a piezoelectric actuator is used in a closed-loop control to compensate for air gap variations by modifying the thickness of the pad. Moreover, in other solutions, piezoelectric actuators were also used to control the air flow exhausted/supplied from/to the bearing [21–23].

Despite their higher effectiveness, active compensation solutions are still too expensive to be integrated into most current industrial applications. By contrast, passive compensation solutions, notwithstanding their lower dynamics and effectiveness, represent an acceptable and cheap alternative. In fact, it was found that, if suitably selected, compliant elements and pneumatic valves can obtain higher and even quasi-static infinite stiffness over 20% of the bearing load range [24]. Regarding passive compensation solutions, Newgard and Kiang [25] proposed the use of elastic orifices to regulate the air flow supplied to the clearance in order to compensate for air gap variations. Another valuable solution proposed by different authors [24,26,27] consists of the use of a bearing with convergent and compliant gap geometries, effecting a passive conicity control. Chen and Lin [28] proposed an aerostatic pad with a disk spring compensator to increase the bearing damping. Ghodsiyeh et al. [29,30] and Colombo et al. [31,32] proposed a novel prototype of an aerostatic pad controlled by means of a diaphragm valve. It was found that this kind of passive compensation makes it possible to obtain quasi-static infinite stiffness and high damping in a low frequency range. Moreover, using pneumatic valves represents a low-cost solution characterized by ease of integration and simple construction. In [33], a prototype of a passively compensated pad was also proposed where a differential diaphragm valve was used.

This paper presents a procedure to suitably design aerostatic pads controlled by diaphragm valves depending on the desired air gap height at which the system has to work. The design procedure is described by considering a simple geometry of the pad, i.e., a circular and centrally fed aerostatic pad. Results demonstrate that this design procedure makes it possible to significantly increase the stiffness of the pad at the desired air gap height. Moreover, it shows how the performance of the compensated pad varies depending on the supply pressure, the ratio between the valve and pad supply hole diameters and the size of the pad.

## 2. The Compensated Pad

### 2.1. Description and Functioning

Figures 1–3 show the schemes of the diaphragm valve and the aerostatic pad considered in the proposed design procedure. As can be seen, the valve is composed of a main body, moving nozzle, upper body, micrometric screw, Belleville washer and a lower body. The nozzle is mounted coaxially with respect to the vertical hole drilled in the main body of the valve and it has an orifice of diameter $d_p$. The presence of the Belleville washer and the micrometric screw makes it possible to regulate its relative initial position (in the absence of supply pressure) ($x_0$) with respect to a diaphragm of diameter $D_m$ and thickness $s$, which is clamped between the main and the lower body of the valve through an O-ring (see Figure 2a). As shown in Figure 2a,b, the initial position (in the absence of supply pressure) of the nozzle can be positive ($x_0 > 0$) when it is not in contact with the diaphragm or negative ($x_0 < 0$) when the nozzle imposes a preload on the diaphragm. When the pressure in the valve chamber $p_1$ increases, the diaphragm deflects, thus increasing the nozzle–diaphragm distance of $x = x_0 + x_v$. Figure 3 shows a scheme of the circular and centrally fed aerostatic pad that is considered for the proposed design procedure.

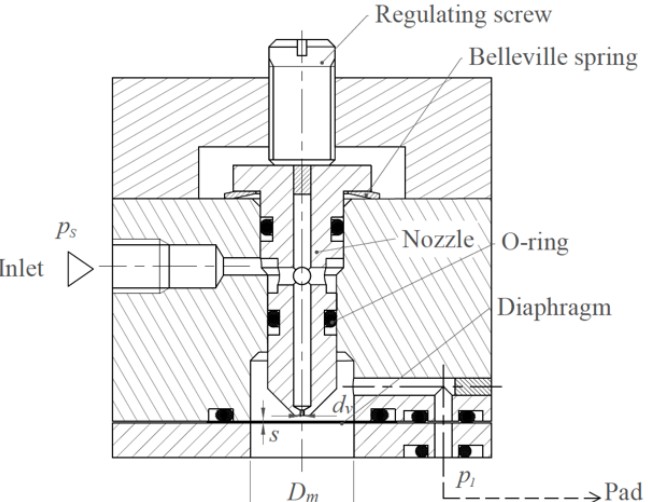

**Figure 1.** Scheme of the diaphragm valve.

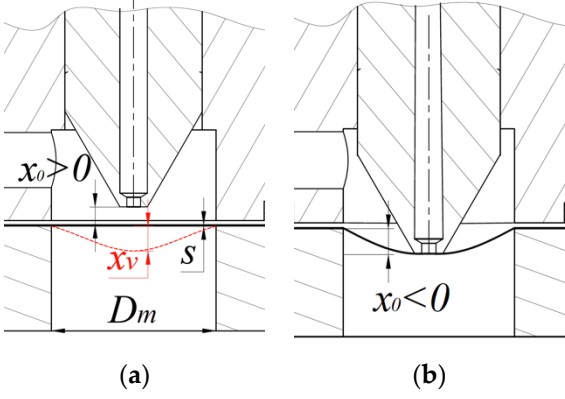

**Figure 2.** (**a**) Configuration for $x_0 > 0$. (**b**) Configuration for $x_0 < 0$.

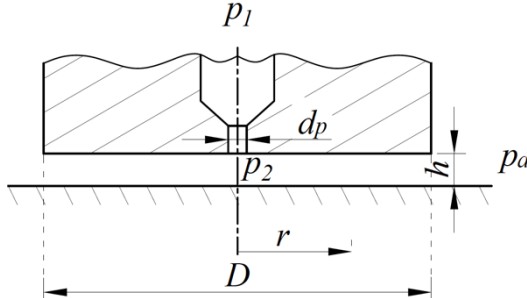

**Figure 3.** Scheme of the aerostatic pad.

The pad has an outer diameter $D$ and a supply hole of diameter $d_p$. The supply pressure of the pad $p_1$ is provided through the outlet of the valve. Once the compressed air reaches the air pad, it passes through the supply hole and then is exhausted from the air gap. During its passage, the air pressure reduces, firstly, to $p_2$ due the pressure drop that occurs at the curtain area under the supply hole and, secondly, it gradually reduces up to ambient pressure $p_a$ at the outer edge of the pad.

The compensated pad is modeled as a lumped pneumatic circuit composed of capacitances and resistances, as shown in Figure 4.

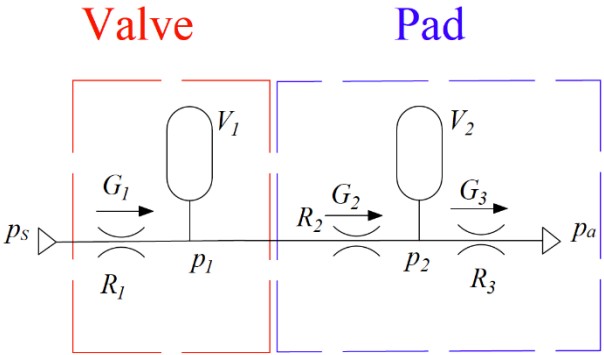

**Figure 4.** Lumped pneumatic circuit of the compensated pad.

where $p_s$ is the supply pressure of the system, $R_1$, $R_2$ and $R_3$ are the lumped resistances related to the valve, the pad and the air gap. $G_1$, $G_2$ and $G_3$ are the correspondent air mass flow rates that are computed as:

$$G_1 = c_{d_1} \frac{p_s}{\sqrt{R_g T_s}} \pi d_v x \, \varphi\left(\frac{p_1}{p_s}\right)$$

$$\varphi\left(\frac{p_1}{p_s}\right) = \begin{cases} \left[\left(\frac{p_1}{p_s}\right)^{2/k} - \left(\frac{p_1}{p_s}\right)^{(k+1)/k}\right]^{1/2} & , \ \frac{p_1}{p_s} < \left(\frac{2}{k+1}\right)^{k/(k-1)} \\ \left[\left(\frac{2}{k+1}\right)^{2/(k-1)} - \left(\frac{2}{k+1}\right)^{(k+1)/(k-1)}\right]^{1/2} & , \ \frac{p_1}{p_s} \geq \left(\frac{2}{k+1}\right)^{k/(k-1)} \end{cases} \quad (1)$$

$$c_{d_1} = 1.05\left(1 - 0.3e^{-0.005 \, Re_1}\right)$$

$$Re_1 = \frac{G_1}{\pi d_v \mu}$$

$$G_2 = c_{d_2} \frac{p_1}{\sqrt{R_g T_1}} \pi d_p h \, \varphi\left(\frac{p_2}{p_1}\right)$$

$$\varphi\left(\frac{p_2}{p_1}\right) = \begin{cases} \left[\left(\frac{p_2}{p_1}\right)^{2/k} - \left(\frac{p_2}{p_1}\right)^{(k+1)/k}\right]^{1/2} & , \ \frac{p_2}{p_1} < \left(\frac{2}{k+1}\right)^{k/(k-1)} \\ \left[\left(\frac{2}{k+1}\right)^{2/(k-1)} - \left(\frac{2}{k+1}\right)^{(k+1)/(k-1)}\right]^{1/2} & , \ \frac{p_2}{p_1} \geq \left(\frac{2}{k+1}\right)^{k/(k-1)} \end{cases} \quad (2)$$

$$c_{d_2} = 1.05\left(1 - 0.3e^{-0.005 \, Re_2}\right)$$

$$Re_2 = \frac{G_2}{\pi d_p \mu}$$

$$G_3 = \frac{\pi h^3 \left(p_2^2 - p_a^2\right)}{12\mu R_g T_2 \cdot ln\left(\frac{D}{d_p}\right)} \tag{3}$$

where $k$, $\mu$ and $R_g$ are the specific heat ratio, dynamic viscosity and gas constant of the air. $T_s$ is the supply temperature of the air, $c_{d_i}$ ($i = 1$, 2) are the discharge coefficients related to the supply holes of the nozzle and the pad (for further details, readers can refer to [34]). The expressions of the air mass flow rates $G_j$ ($j = 1$, 2 , 3) are obtained by considering isentropic expansions through the nozzle and the supply hole and isothermal conditions in the air gap. It is worth noting that because of the large heat exchange between the metallic part of the system and the small volumes occupied by the fluid, all the temperatures are considered equal to the ambient temperature ($T_s = T_1 = T_2 = T_a$). The load capacity of the compensated pad can be obtained by integrating the expression of the air gap pressure distribution (for further details, see [35]):

$$p(r) = p_2 \sqrt{1 - \left(1 - \frac{p_a^2}{p_2^2}\right) \frac{ln\left(\frac{2r}{d_p}\right)}{ln\left(\frac{D}{d_p}\right)}} \tag{4}$$

$$F_P = p_2 \frac{\pi d_p^2}{4} \sqrt{\frac{\pi A}{8}} e^{\frac{2}{A}} \left[erf\left(\sqrt{\frac{2}{A}}\right) - erf\left(\sqrt{\frac{2}{A}} \cdot \frac{p_a}{p_2}\right)\right] \quad A = \frac{\left(1 - \frac{p_a^2}{p_2^2}\right)}{ln\left(\frac{D}{d_p}\right)} \tag{5}$$

The considered compensation method is based on the fact that, for each value of the external load applied upon a pad ($F^{ext} = F_p$), there exists a supply pressure $p_1$ that makes it possible to keep constant a desired air gap height $h^*$. In order to clarify the working principle of the compensation method, Figure 5a,b compare the operating principles of a conventional to that of the considered compensated pad. These figures compare how a conventional pad (Figure 5a) and a compensated pad (Figure 5b) work when the external load $F$ increases. Here, continuous lines represent the air flow $G_2$ supplied by the supply holes of the pad for a given supply pressure $p_1$ and a variable downstream pressure $p_2$. Similarly, dotted lines represent the air flow $G_3$ exhausted from an air gap with a constant height. Given an initial condition ($h_0$, $G_0 = G_2 = G_3$, $F_0$), in a conventional pad, the air flow $G$ and the air gap height $h$ decrease as the applied load $F$ increases (see Figure 5a). This is because the operating principle of the pad depends on the air flow supplied by its supply holes. The idea of the proposed compensation method is to exploit the presence of the diaphragm valve to compensate for these air gap height variations by increasing the supply pressure of the bearing. As can be seen from Figure 5b, if the diaphragm valve were able to suitably modify the supply pressure of the pad in accordance with external load variations, the compensated pad would work at a constant air gap height, thus providing quasi-static infinite stiffness.

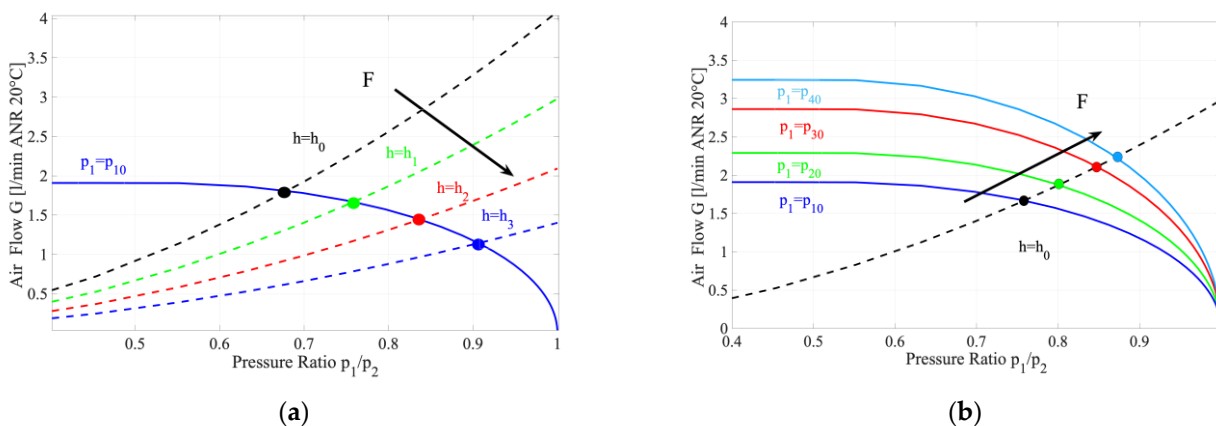

**Figure 5.** (**a**) Working conditions of a conventional pad. (**b**) Working conditions of a compensated pad.

The aim of this work is to show how to design the compensated pad (the valve and the pad) in order to obtain the highest stiffness in a neighborhood of a desired air gap height $h^*$.

### 2.2. Design Procedure

Once a desired air gap height $h^*$ is decided, the most important parameters that have to be defined to maximize the stiffness of the pad are:

- the stiffness of the diaphragm ($k_m$);
- the initial position of the nozzle with respect to the diaphragm ($x_0$).

The first step in defining the values of these parameters is to compute the ideal values of the supply pressure of the pad $p_1{}^{id}$ that make it possible to obtain a constant air gap height $h^*$. These pressure values can be easily computed by applying the continuity equation at the air gap entrance:

$$\overbrace{c_{d_2}(Re, h^*)\frac{p_1{}^{id}}{\sqrt{R_g T_1}}\pi d_p h^*\ \varphi\left(\frac{p_2}{p_1{}^{id}}\right)}^{G_2} - \overbrace{\frac{\pi\ h^{*3}\left(p_2^2 - p_a^2\right)}{12\mu R_g T_s\cdot ln\left(\frac{d}{d_p}\right)}}^{G_3} = 0 \tag{6}$$

Based on the assumption that, in most cases, external loads applied upon the pad $F$ are proportional with respect to the pressure downstream of its supply holes $p_2$, given the external loads and the desired air gap height $h^*$, Equation (6) can be used to compute the corresponding $p_1^{id}$. Being nonlinear, Equation (6) must be solved iteratively through the Regula Falsi Method. It is worth noting that discharge coefficients can be also considered as a function of the Reynolds number $Re$ and the considered air gap height $h^*$ using different formulations [34] (as shown in Equations (1) and (2)).

Once $p_1^{id}$ has been computed for each value of the applied load $F$, it is necessary to limit the maximum value of $p_1^{id}$ considering the maximum supply pressure that can be provided to the valve. In these instances, the maximum value was taken as equal to 0.7 MPa. This choice was made considering the supply pressures that are normally used in conventional gas bearing applications ([36], see page 228). Figure 6 shows the trend of the ideal and effective (considering the limit of 0.7 MPa) supply pressure $p_1$ as a function of the applied load (computed through Equation (5)). As can be seen, the range of compensation reduces as the maximum supply pressure is reduced.

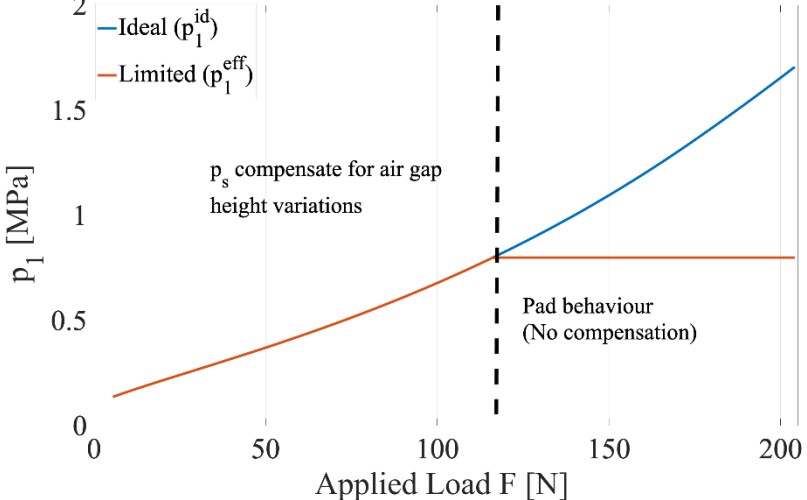

**Figure 6.** Trend of $p_1^{id}$ and $p_1^{eff}$ versus the applied load $F^{ext}$.

At this stage, given that

- the air mass flow rate $G = G_3$;
- the effective supply pressure that the valve must supply to the pad $p_1^{eff}$,

it is possible to compute the correspondent valve opening law: the distance between the nozzle and the diaphragm $x$. Similarly to the previous step, this can be done by imposing the continuity equation at the valve chamber.

$$x^{id} = \frac{G_3 \sqrt{R_g T_s}}{\pi d_v p_s \, \varphi\left(\frac{p_1^{eff}}{p_s}\right)} \tag{7}$$

The only difference from the previous step is that the dimension of the nozzle diameter is unknown. In order to define a suitable value of this diameter $d_v$, $x$ were computed considering different ratios of $d_v/d_p$: 0.5, 0. 25 and 0.1.

Figure 7 shows the trend of the ideal distance $x^{id}$ versus the pressure in the valve chamber for different $d_v/d_p$ ratios. Here, it is possible to see that adopting lower ratios of $d_v/d_p$ results in more compliant diaphragms with a more nonlinear behavior. In view of this, a better choice would be to consider a ratio as large as possible to reduce the dimensions and the nonlinear behavior of the valve diaphragm. On the other hand, the smaller the diameter ratio, the more effective the compensating action of the valve. This can be argued considering the curtain areas downstream of the valve nozzle and the supply hole of the pad:

$$\pi d_p h < \pi d_v x \qquad \text{Pad behaviour (Case 1)}$$
$$\pi d_p h > \pi d_v x \qquad \text{Valve compensation (Case 2)}$$

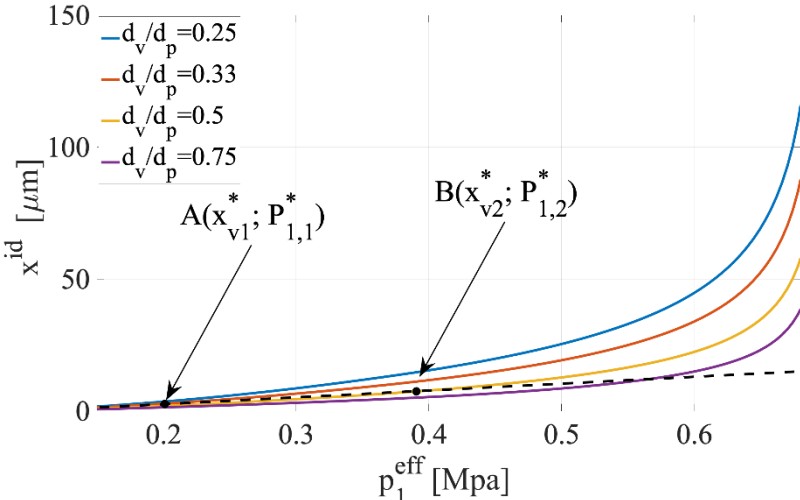

**Figure 7.** Trend of the ideal distance $x^{id}$ versus the pressure in the valve chamber for different $d_v/d_p$ ratios.

Depending on which of these curtain areas is the smaller one, the system behavior is governed by the pad (Case 1) or the compensating action of the valve (Case 2). After these considerations, the ratio $d_v/d_p$ was taken as equal to 0.5 ($d_p = 1$ mm and $d_v = 0.5$ mm). At this point, on the basis of the curve obtained in Figure 7, it is necessary to approximate the ideal curve representing the valve opening law with a straight line. The better the straight line approximates the ideal curve, the closer the real behavior of the system will be to the ideal one. In view of this, the line was taken between the ideal curve at pressures equal to 0.2 MPa and $\left(\frac{2}{k+1}\right)^{k/(k-1)} p_s$.

Once these two points A $(x_{v,1}^*; p_{1,1}^*)$ and B $(x_{v,2}^*; p_{1,2}^*)$ are defined, by suitably manipulating the equation of the line, it is possible to obtain the values of the corresponding diaphragm stiffness $k_m$ and its initial distance from the nozzle $x_0$:

$$x_0 = x_{v,1}^* - \left( \frac{x_{v,2}^* - x_{v,1}^*}{p_{1,1}^* - p_{1,2}^*} \right) \times p_{1,2}^* \tag{8a}$$

$$k_m = \frac{\pi D_m^2}{4 \left( \frac{x_{v,2}^* - x_{v,1}^*}{p_{1,2}^* - p_{1,1}^*} \right)} \tag{8b}$$

where $D_m$ = 3 mm is the effective diameter of the diaphragm (the diameter corresponding to the active surface of the diaphragm) that, in order to reduce the valve dimensions, was taken as 1 mm larger than the outer diameter of the nozzle.

### 2.3. Computational Algorithm to Obtain the Bearing Features

When the parameters that characterize the behavior of the compensated pad were defined, namely $k_m$, $x_0$, $d_v$, $d_p$, $p_s$, $h^*$, the static performance was obtained through the lumped model depicted in Figure 4. Here, the equations related to the analytical solution of a circular and centrally fed pad were used to compute the load capacity (Equation (5)) and the air flow (Equation (3)) of the pad. The presence of the lumped volumes $V_1$ and $V_2$ was necessary to iteratively calculate the correspondent pressure $p_1$ and $p_2$. The structure of the algorithm to compute the static curves of the compensated pad is similar to that adopted in [31] and it has been implemented in the Matlab environment. The algorithm consists of two main parts. The first part is necessary to compute an initial static solution of the problem $(h_0, F_{p_0}, G_0, P_{0_0}, P_{1_0}, P_{2_0})$, where the load capacity $F_{p_0}$ and the air mass flow rate $G_0$ of the pad are computed by considering the air gap height $h_0$ as an input parameter of the model. Once the convergence conditions on the load capacity and flow rate are simultaneously satisfied, the resulting physical parameters $h_0, F_{p_0}, G_0, P_{0_0}, P_{1_0}, P_{2_0}$ are used as input variables in the second main part of the algorithm. In this second step, the air gap height values that guarantee the equilibrium of the pad are iteratively computed by simulating the application of a step force $\Delta F$ (in the time domain). Given the initial static condition $(h_0, F_{p_0}, G_0, P_{0_0}, P_{1_0}, P_{2_0})$ and the selected step force $\Delta F$, the external load $F^{ext}$ acting on the pad is computed as:

$$F^{ext}{}_i = F_{p_0} + i \cdot \Delta F \tag{9}$$

where, $i$ is the number of iterative steps that have been already solved ($i = 0, 1, 2, \dots, i_{max}$). In each iterative step ($i$), the equilibrium air gap height is computed by solving the equilibrium equation of the pad (Equation (10)) through Euler's explicit method:

$$F^{ext} = F_p - M\ddot{h} \tag{10}$$

where $M$ is the moving mass of the pad. Once the equilibrium is reached ($M\ddot{h} \approx 0$ and the convergence conditions are satisfied), the obtained results are used as the new initial condition, and the new external load $F^{ext}$ is obtained from the previous one by adding a further step force $\Delta F$ (see Equation (9)). As discussed in [31], it is necessary to use this kind of algorithm since the compensating action of the valve renders the load capacity $F_p$ and the air mass flow rate $G$ non-injective functions of the air gap height $h$ (there is more than one load capacity and air mass flow rate for the same air gap height value).

## 3. Sensitivity Analysis

The lumped model and the proposed design procedure are used to perform a sensitivity analysis aimed at investigating the influence of the pad size (diameter $D$), the ratio between the pad and valve supply holes ($d_v/d_p$), the desired air gap height ($h^*$) and valve

supply pressure ($p_s$). Figure 8 shows a scheme that summarizes what has been done in the present work. Firstly, the design procedure and the lumped model were applied to a reference case where the input parameters were chosen by considering the supply pressure ($p_s$ = 0.7 MPa) and bearing size ($D$ = 40 mm) that are conventionally used in gas bearing applications and supply hole diameters that can favor the compensation action of the valve and the manufacturing of the valve nozzle ($d_v$ = 0.5 mm and $d_p$ = 1 mm). In fact, taking large supply hole diameters of the pad makes it possible to obtain a compensated system whose behavior is mainly governed by the valve nozzle and the air gap. Then, this analysis was extended to evaluate the effect of the influence of size of the pad, the ratio between the pad and valve supply holes, the desired air gap height and valve supply pressure.

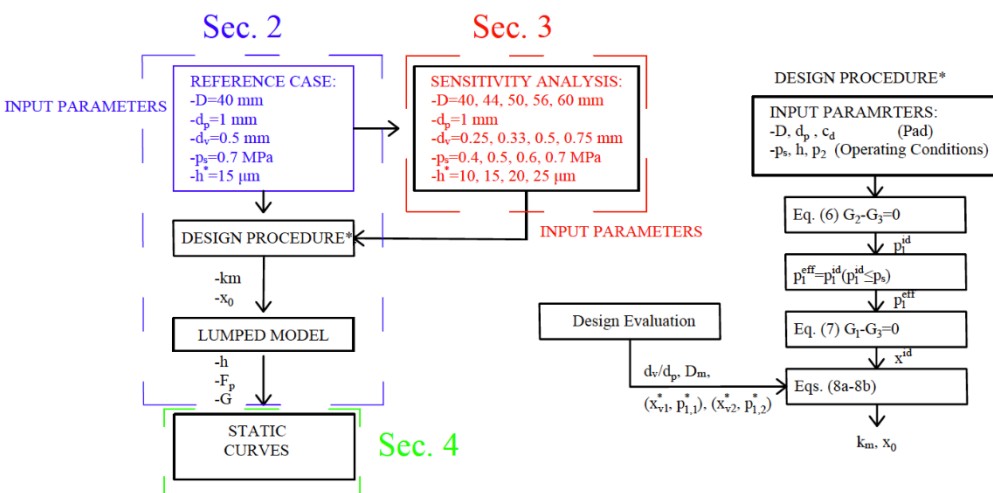

**Figure 8.** Flow charts summarizing the steps of the proposed design procedure and sensitivity analysis.

## 4. Results and Discussion

Figures 9–21 show the results of the performed sensitivity analysis.

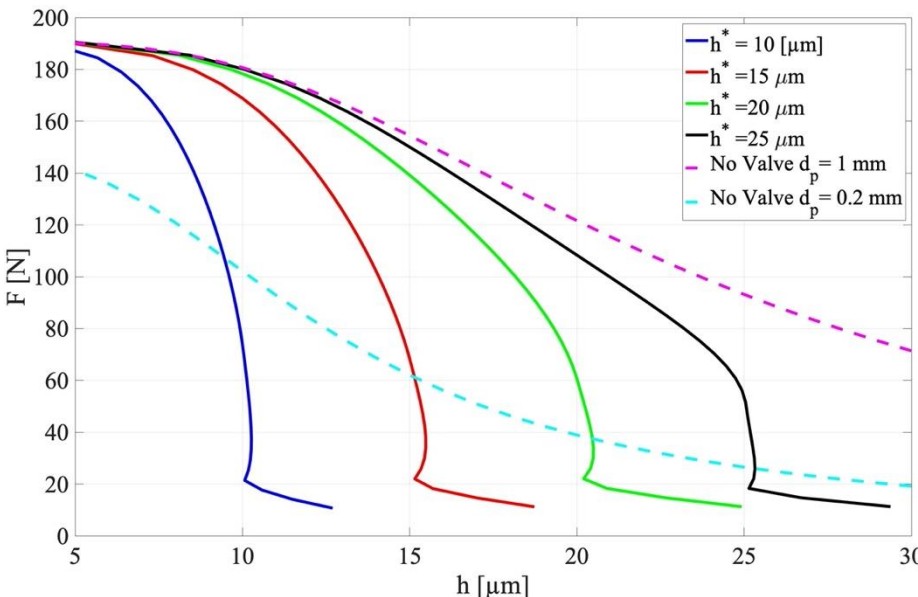

**Figure 9.** Comparison among the load capacities of the two benchmarks and compensated pad designed considering different desired air gap heights $h^*$, by considering the other parameters as constant: $D$ = 40 mm, $d_p$ = 1 mm, $d_v$ = 0.5 mm and $p_s$ = 0.7 MPa.

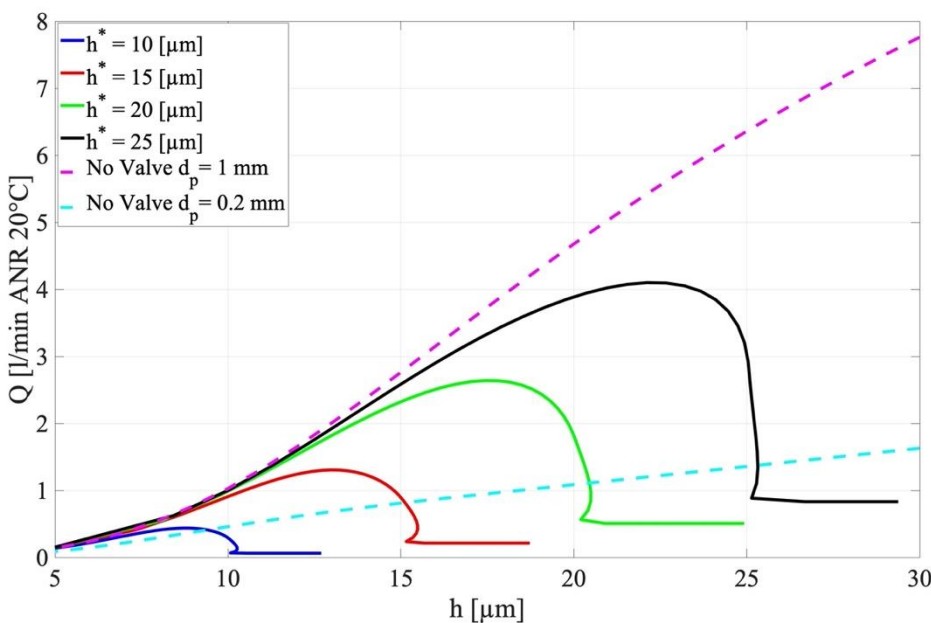

**Figure 10.** Comparison among the air consumptions of the two benchmarks and compensated pad designed considering different desired air gap heights $h^*$, by considering the other parameters as constant: $D = 40$ mm, $d_p = 1$ mm, $d_v = 0.5$ mm and $p_s = 0.7$ MPa.

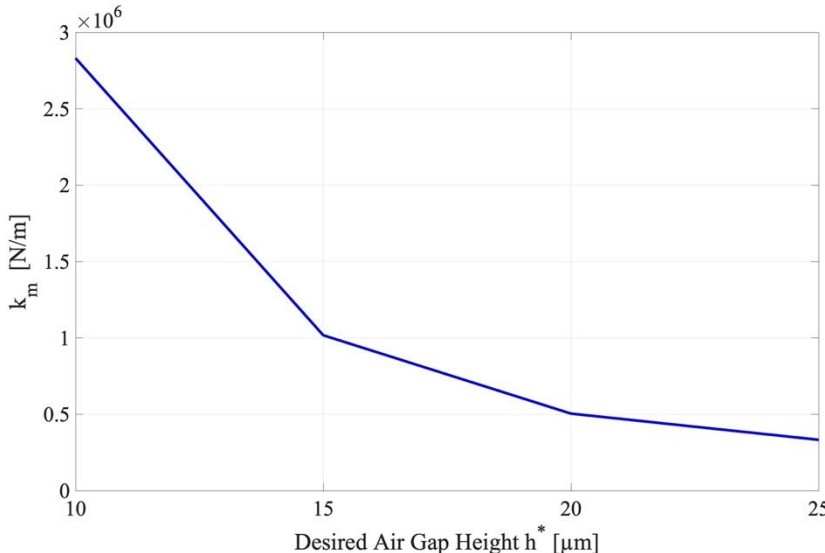

**Figure 11.** Trend of the optimal diaphragm stiffness expressed as a function of the desired air gap height $h^*$, by considering the other parameters as constant: $D = 40$ mm, $d_p = 1$ mm, $d_v = 0.5$ mm and $p_s = 0.7$ MPa.

### 4.1. Effect of the Selected Air Gap Height

Figures 9 and 10 show the load capacity and the air consumption of different compensated pads designed to work at different air gap heights $h^*$ = 10, 15, 20 and 25 μm, whereas the other input parameters are chosen equal to those of the reference case indicated in Figure 8 ($p_s = 0.7$ MPa, $D = 40$ mm, $d_v = 0.5$ mm and $d_p = 1$ mm). The characteristic curves of the non-compensated pad and a "more conventional" pad ($d_p = 0.2$ mm) were used as benchmark curves to better figure out the efficiency of the proposed method. It is possible to see that the stiffness of the compensated pad is much higher than those of the benchmark curves over almost the entire analyzed load range and for all the desired air gap heights $h^*$ (Figure 9). As regards the air consumption (Figure 10), for the compensated

pad, it increases as the desired air gap height is increased too. However, by considering the common air gap heights used in industrial applications (around 10–15 μm), the air consumption is quite similar or lower than those of the second benchmark ($d_p = 0.2$ mm). Other relevant considerations can be made on the diaphragm stiffness that is required to obtain these compensated pads. As it is possible to see in Figure 11, this stiffness decreases in a nonlinear way as the desired air gap height is decreased.

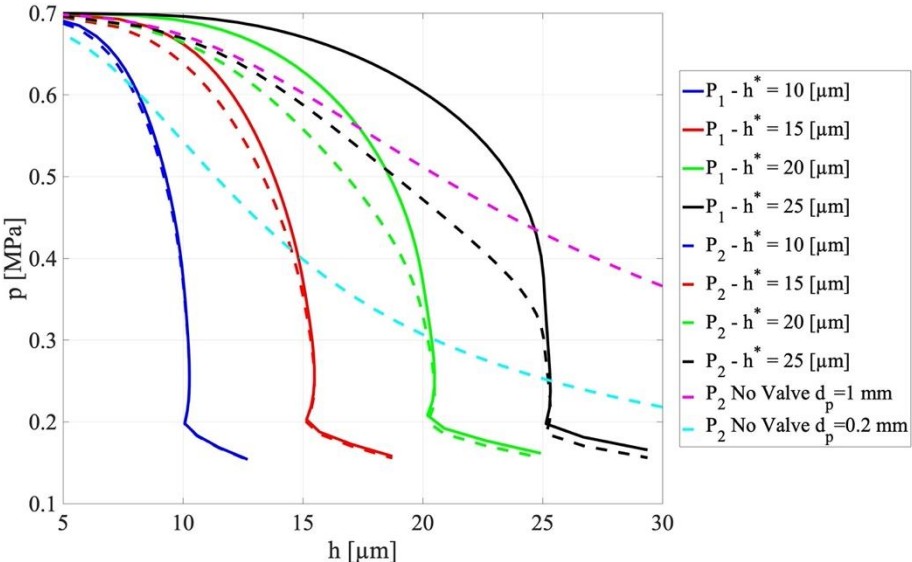

**Figure 12.** Comparison between the pressures upstream and downstream of the orifice of the pad considering different values of the desired air gap height $h^*$, by considering the other parameters as constant: $D = 40$ mm, $d_p = 1$ mm, $d_v = 0.5$ mm and $p_s = 0.7$ MPa.

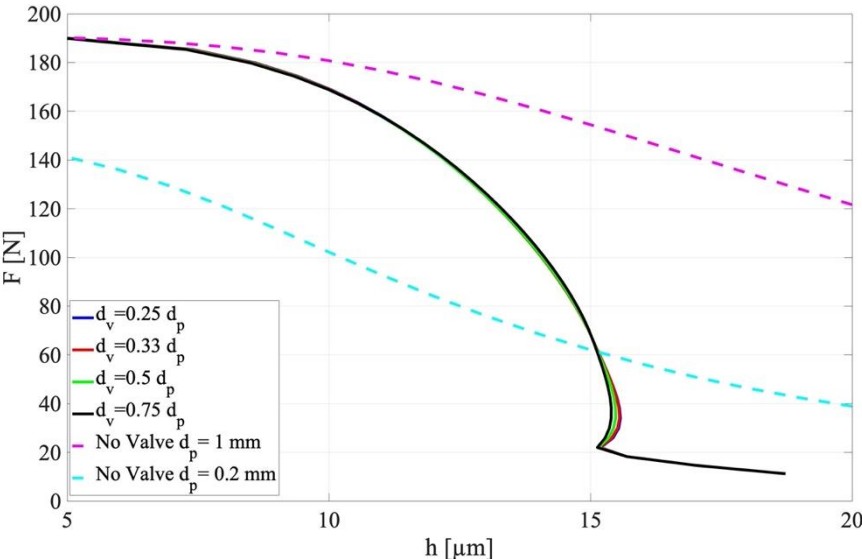

**Figure 13.** Comparison among the load capacities of the two benchmarks and compensated pad designed considering different ratios of the nozzle $d_v$ diameters, by considering the other parameters as constant: $D = 40$ mm, $h^* = 15$ μm and $p_s = 0.7$ MPa.

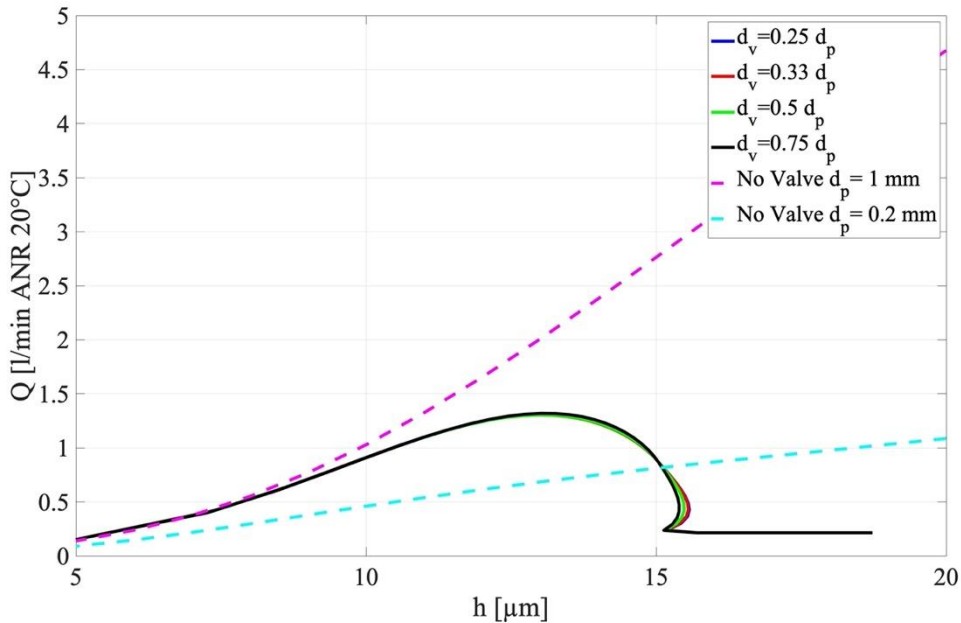

**Figure 14.** Comparison among the air consumptions of the two benchmarks and compensated pad designed considering different ratios of the nozzle $d_v$ and orifice $d_p$ diameters, by considering the other parameters as constant: $D = 40$ mm, $h^* = 15$ μm and $p_s = 0.7$ MPa.

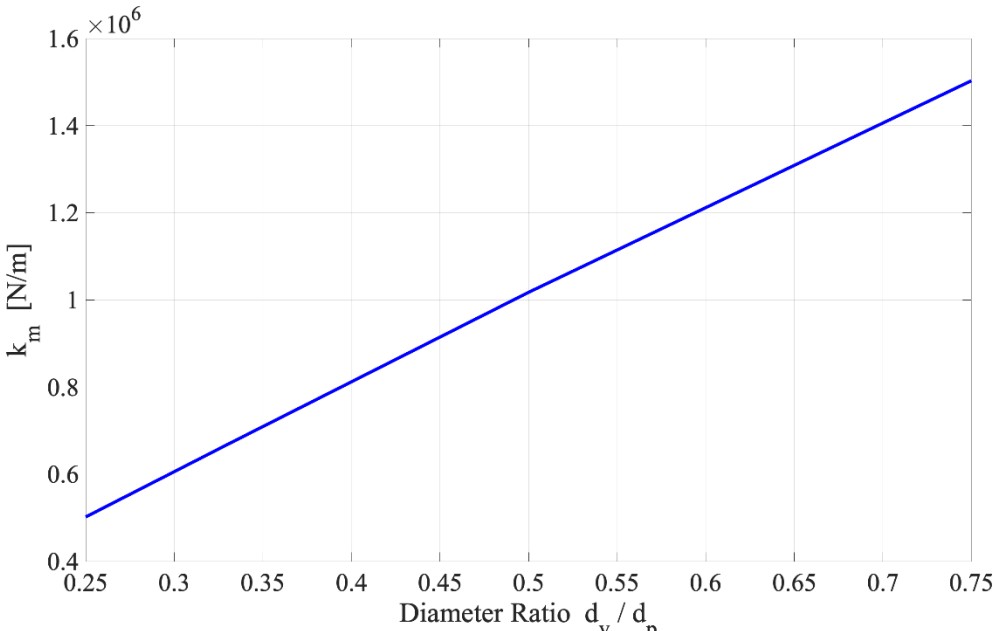

**Figure 15.** Trend of the optimal diaphragm stiffness expressed as a function of the diameter ratio $d_v/d_p$ and orifice $d_p$ diameters, by considering the other parameters as constant: $D = 40$ mm, $h^* = 15$ μm and $p_s = 0.7$ MPa.

Further insight into the functioning of the compensated pad can be obtained by considering the pressure drop occurring downstream of the supply hole of the pad. From Figure 12, it is clear that, thanks to the valve regulation, close to the desired air gap height, the pressure drop downstream of the supply hole of the pad is almost zero (the presence of this hole has no effect on the behavior of the pad). This effect becomes more relevant as the desired air gap height is decreased.

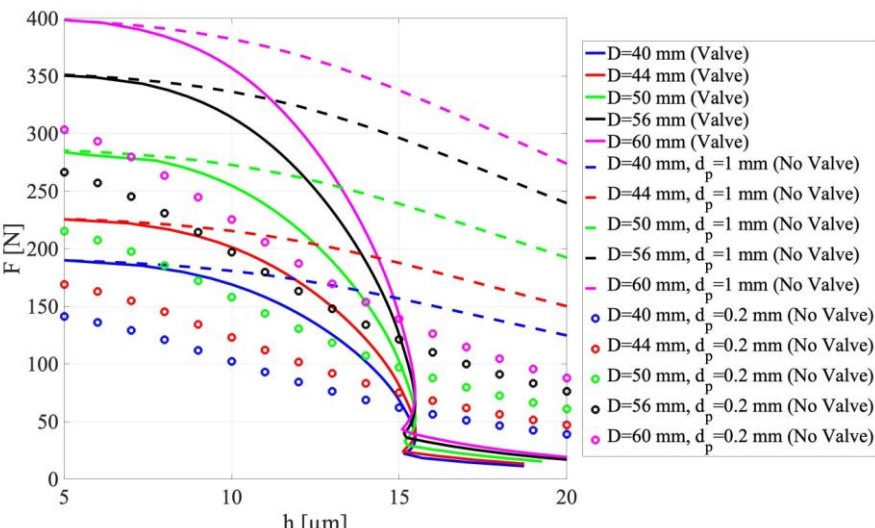

**Figure 16.** Comparison among the load capacities of the two benchmarks and compensated pad designed considering different outer diameters of the circular pad $D$, by considering the other parameters as constant: $d_p = 1$ mm, $d_v = 0.5$ mm, $h^* = 15$ µm and $p_s = 0.7$ MPa.

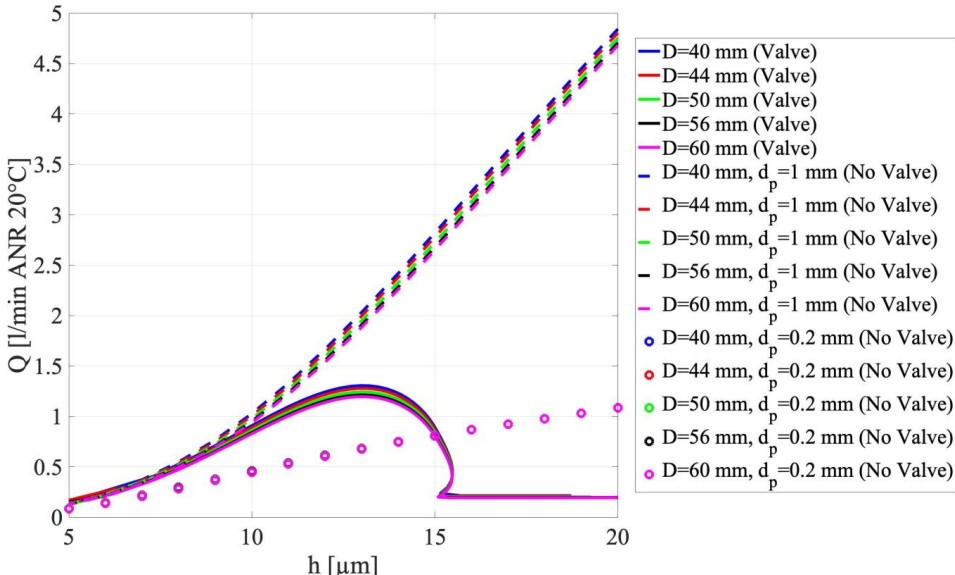

**Figure 17.** Comparison among air consumption of the two benchmarks and compensated pad designed considering different outer diameters of the circular pad $D$, by considering the other parameters as constant: $d_p = 1$ mm, $d_v = 0.5$ mm, $h^* = 15$ µm and $p_s = 0.7$ MPa.

### 4.2. Effect of the Diameter Ratio

Figures 13–15 report the results from the simulations aimed at investigating the effect of varying the supply hole of the valve $d_v$ keeping constant the supply hole of the pad $d_p = 1$ mm. This because relatively large diameters for the supply hole of the pad are essential to obtain higher performance from this kind of compensation system. As it is possible to see from Figures 13 and 14, varying the diameter of the supply hole of the valve does not affect the performance of the compensation system. This can be easily understood by considering that, in this instance, the reduction in the conductance of the valve supply hole (due to the lower diameter) is compensated by an increase in the diaphragm stiffness (lower values of $x$).

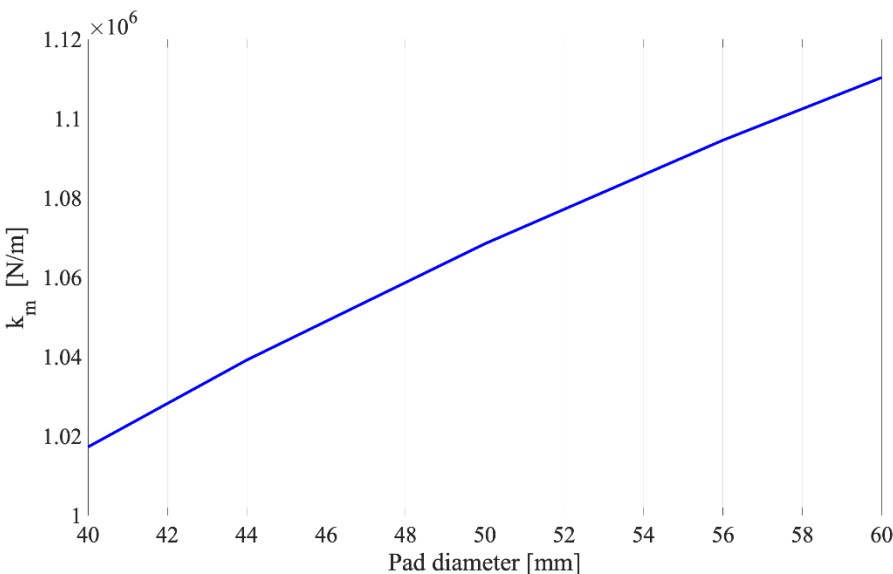

**Figure 18.** Trend of the optimal diaphragm stiffness expressed as a function of the outer diameter of the pad *D*, by considering the other parameters as constant: $d_p = 1$ mm, $d_v = 0.5$ mm, $h^* = 15$ μm and $p_s = 0.7$ MPa.

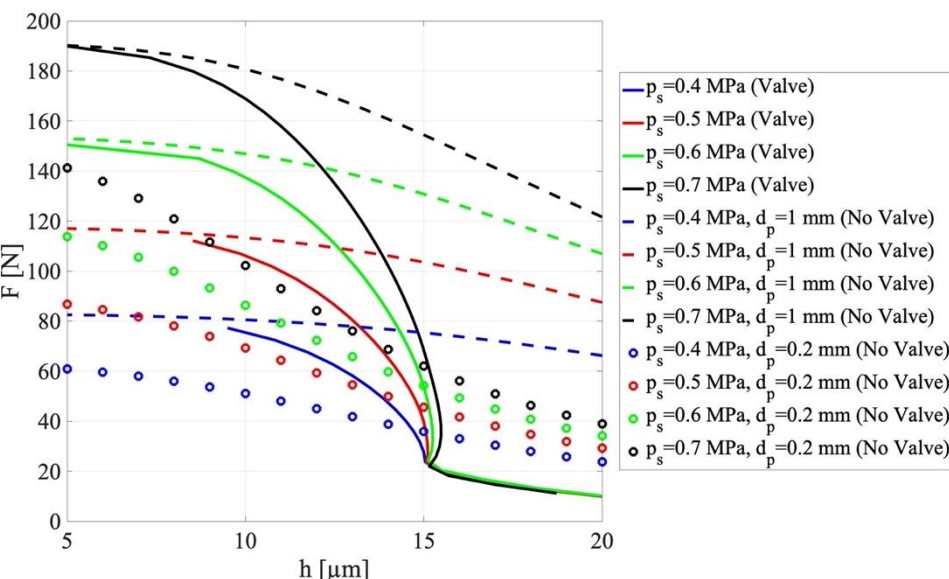

**Figure 19.** Comparison among the load capacities of the two benchmarks and compensated pad designed considering different values of the supply pressure $p_s$, by considering the other parameters as constant: $d_p = 1$ mm, $d_v = 0.5$ mm, $h^* = 15$ μm and $D = 40$ mm.

### 4.3. Effect of the Pad Size

The influence of increasing the pad size was investigated by considering the performance of different compensated pads with outer diameters that are larger than the reference case of 10, 25, 40 and 50%. As can be seen from Figure 16, the region of maximum stiffness (around the desired air gap height $h^*$) of the pad increases proportionally with respect to the size of the pad. Conversely, the air consumption is almost the same considering this increment in the pad size (Figure 17). Additionally, in this case, to obtain higher performance, the stiffness of the diaphragm has to suitably increase with the diameter of the pad. Hence, increasing the size of the pad integrated does not reduce the effectiveness of the method when this size variation is compensated by a suitable variation in the diaphragm stiffness (Figure 18).

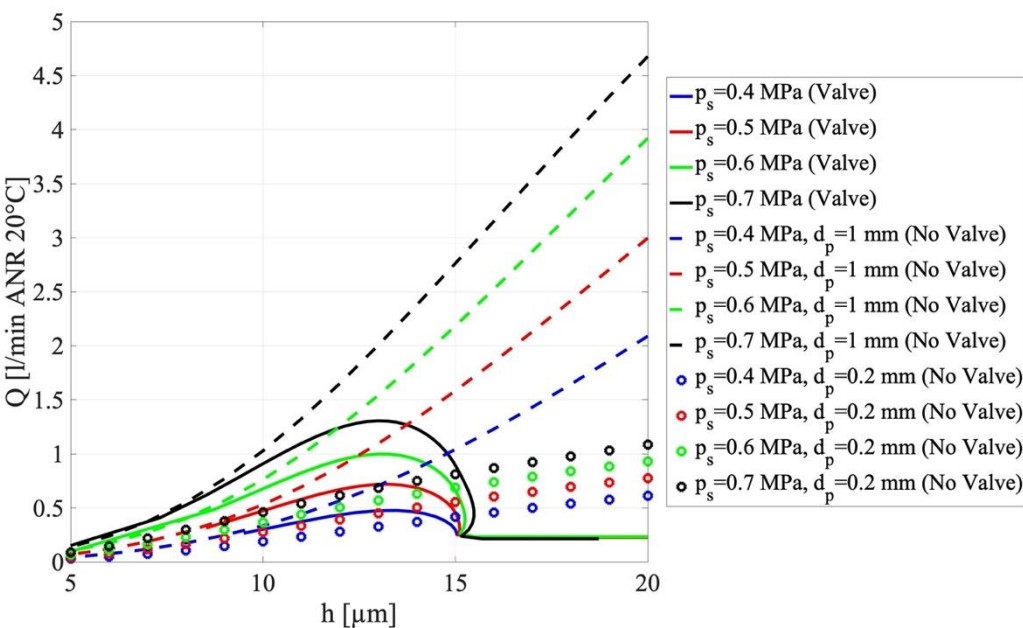

**Figure 20.** Comparison among the air consumptions of the two benchmarks and compensated pad designed considering different values of supply pressure $p_s$, by considering the other parameters as constant: $d_p = 1$ mm, $d_v = 0.5$ mm, $h^* = 15$ μm and $D = 40$ mm.

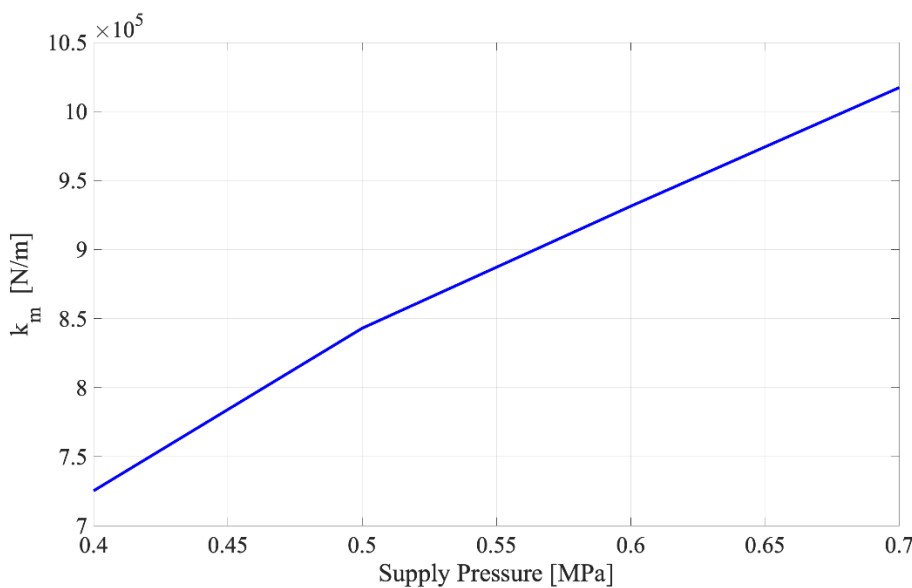

**Figure 21.** Trend of the optimal diaphragm stiffness expressed as a function of the supply pressure $p_s$, by considering the other parameters as constant: $d_p = 1$ mm, $d_v = 0.5$ mm, $h^* = 15$ μm and $D = 40$ mm.

### 4.4. Effect of the Supply Pressure

Figures 19 and 20 show the results of the simulations aimed at investigating the effect of reducing the supply pressure of the compensated pad (reference case, see Figure 8). As expected, reducing the supply pressure of the compensated pad reduces both the maximum load capacity and the region of maximum stiffness (Figure 19). As regards air consumption (Figure 20), it obviously reduces with the supply pressure and it is always of the same order of the second benchmark ($d_p = 0.2$ mm). As can be seen from Figure 21, the stiffness of the membrane reduces as the supply pressure is reduced.

## 5. Conclusions

This paper presents a design procedure for a circular and centrally fed aerostatic pad controlled by a diaphragm valve. As demonstrated by previous numerical and experimental works [29–31,33], this kind of compensation method makes it possible to significantly increase the static stiffness of air pads up to quasi-static infinite stiffness. Here, a design procedure is proposed to suitably select the geometry of this kind of passively compensated pad depending on the values of a desired air gap height at which the system has to work. The numerical results demonstrate that, thanks to this design procedure, it is always possible to obtain a quasi-infinite stiffness in a neighbor of the desired air gap height. Moreover, a sensitivity analysis was performed to study the effect of varying the desired air gap height, supply pressure, diameter of the valve supply hole and the pad size. It was found that the proposed design procedure is effective in the presence of all these variations. In particular, the most relevant results are that:

- reducing the value of the desired air gap height globally increases the stiffness of the system along with the compensation range;
- increasing the outer radius of the integrated pad or modifying the supply hole diameter of the valve does not reduce the effectiveness of the compensation method but it results in a different value of the required diaphragm stiffness;
- given the desired air gap height, the extent of the compensation range is almost proportional to the supply pressure.

Future works will focus on numerical simulation by considering bearings with multiple orifices and experimental tests to validate the preliminary results obtained herein.

**Author Contributions:** Conceptualization, L.L., T.R., V.V., F.C., A.T.; Data curation, L.L.; formal analysis, L.L.; funding acquisition, T.R.; Investigation, L.L., T.R., V.V., F.C., A.T.; methodology, L.L., T.R., V.V., F.C., A.T.; project administration, T.R., V.V.; software, L.L.; supervision, T.R., V.V., F.C., A.T. Validation, T.R., V.V., F.C., A.T.; writing—original draft, L.L.; writing—review and editing, L.L.;. All authors have read and agreed to the published version of the manuscript.

**Funding:** This research received no external funding.

**Conflicts of Interest:** Authors declare no conflict of interest.

### Nomenclature

| | |
|---|---|
| $D$ | Outer diameter of the pad (m) |
| $D_m$ | Diameter of the valve membrane (m) |
| $F_p$ | Load capacity of the pad (N) |
| $F^{ext}$ | Applied load (N) |
| $G_1$ | Air flow rate through the valve nozzle (kg/s) |
| $G_2$ | Air flow rate through the pad orifice(kg/s) |
| $G_3$ | Air flow rate through the air gap (kg/s) |
| $M$ | Moving mass of the pad (kg) |
| $Q$ | Air flow rate (l/min ANR 20 °C) |
| $R_g$ | Air constant (J/(kg·K)) |
| $R_1$ | Pneumatic resistance of the valve nozzle ((s·Pa)/kg) |
| $R_2$ | Pneumatic resistance of the pad orifice ((s·Pa)/kg) |
| $R_3$ | Pneumatic resistance of the air gap ((s·Pa)/kg) |
| $Re_1$ | Reynolds number of the valve nozzle (-) |
| $Re_2$ | Reynolds number of the pad orifice (-) |
| $T_a$ | Ambient temperature (K) |
| $T_s$ | Supply temperature (K) |
| $T_1$ | Valve chamber temperature (K) |
| $T_2$ | Air gap temperature (K) |

| | |
|---|---|
| $V_1$ | Volume of the valve chamber ($m^3$) |
| $V_2$ | Volume at the air gap inlet ($m^3$) |
| $c_{d_1}$ | Valve nozzle discharge coefficient (-) |
| $c_{d_2}$ | Pad orifice discharge coefficient (-) |
| $d_v$ | Diameter of the valve nozzle (m) |
| $d_p$ | Diameter of the pad orifice (m) |
| $k$ | Ratio of the air specific heats (-) |
| $k_m$ | Diaphragm stiffness (N/m) |
| $h$ | Air gap height (m) |
| $h^*$ | Desired air gap height (m) |
| $p_s$ | Valve supply pressure (Pa) |
| $p_a$ | Ambient pressure (Pa) |
| $p_1$ | Pressure at the valve chamber (Pa) |
| $p_2$ | Pressure at the air gap inlet (Pa) |
| $p_{1,i}^*$ | Supply pressure selected for the diaphragm design (Pa) |
| $p_1^{id}$ | Ideal supply pressure of the pad (Pa) |
| $p_1^{eff}$ | Effective supply pressure of the pad (Pa) |
| $s$ | Diaphragm thickness (m) |
| $x$ | Membrane-Nozzle distance (m) |
| $x^{id}$ | Ideal diaphragm deflection (m) |
| $x_0$ | Initial membrane-nozzle distance (m) |
| $x_v$ | Membrane deflection due to the air pressure (m) |
| $x_{v,i}^*$ | Deflection selected for the diaphragm design (m) |
| $\mu$ | Dynamic viscosity ($Ns/m^2$) |

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
