# Peer review of "Design and Analysis of an Aerostatic Pad Controlled by a Diaphragm Valve"

_lubricants, doi:10.3390/lubricants9050047_

Round 1

Reviewer 1 Report

The manuscript presents a theoretical study of an aerostatic bearing with a novel supply system, controlled through a diaphragm valve. The paper is focused on the design procedure, based on previously published analyses of the same group of scientists.

The subject is undoubtedly a subject of interest for high precision mechanical devices. However, the element of novelty is relatively limited, because:

(a) The solution of diaphragm valves have already been presented by the same authors in a series of 4 recently published papers;

(b) The model was already presented in previous papers belonging to the same scientific group; herein, the equations are synthetically presented. Moreover, one of the reference is not yet published (ref. 14).

(c) The algorithm for the solution of the system of equations is presented elsewhere (ref. [30]) - see lines 209-210.

(d) The numerical results refer only to a particular case.

The TITLE of the paper includes the word "modelling", which is not reflected the content: the paper presents clearly the DESIGN PROCEDURE (stated in the abstract - lines 13 & 14) for the supply pressure system of a thrust, circular pad (note that there is no mention of THRUST BEARINGS). The bases of the model were presented in previously published papers.

At line 84 it is stated that: "This paper presents a design procedure to suitably select the geometry and size of the pad". What is the "geometry of the pad" and where is the method for the selection of the size presented?

Eqs. (7) presents two formulations for the discharge coefficient; however, these formulations (which can be easily found in ref. [33] are already presented in eqs. (1) and (2). Moreover, (eq.7b) is not used, therefore making it  useless for this practical example.

A key element is optimal operation of this supply system is the diaphragm; there is no detail of its elastic characteristic (pressure-deformation). Is it linear?  As long as this paper presents a numerical analysis of a practical case, numerical values of x distances will help the reviewer to understand the numerical results.

Figs. 15,18, & 21, showing the stiffness calculated for the variation of a parameter (orifice diameter ratio, pad diameter and supply pressure) are hard to follow: firstly, the values on y-axes are different, despite having the same variable; secondly, stiffness is a complex function of multiple parameters and for each graph, the constant parameters MUST BE presented. For example, for what values of the supply pressure, pad diameter etc. was the graph from fig. 15  traced? The same requirement stands for the other graphs, which must include the constant values of the parameters.

STYLE

Figs. 2 :  "Configuration FOR x0>0 /x0<0" , instead of "in the presence of", which has no meaning here

line 191 " ...system behaviour IS governed..."

line 222 - where is eq. (16)?

line 278 - "pads with outer diameterS that are larger than the reference case WITH 10, 40 and 50%" (Note that the case d= 50mm is not shown in figs.  16 & 17)

Titles of figures are questionable or even erroneous: some present the name of the parameter of the graphs (Figs. 13-17) other are identical with the x-axis (Figs. 11 & 18) or y-axis titles (Fig. 12).

The legend of figs. 13 & 14 presents Dv and D0 - diameters which cannot be found in the list of notations.

Corrections for the text and figures:

Line 17: "....to almost  50%" is correct.

NOTATIONS

Some notations could be confusing, therefore I would recommend a few changes:

- Subscripts "v" and "p" refer to the "valve nozzle" and "pad orifice" respectively, and the corresponding discharge coefficients have the subscripts 1 and 2; subscripts "1" and "2" refer to  the "valve chamber" and "air gap" (bearing interstice).

- The list of notation misses the mass – M.

REFERENCES

Refs. [20] and [21] are the same.

Refs. [17] and [18] are available in Japanese; the language  should be, at least, specified.

Author Response

Dear Reviewer,

Thank you very much for your valuable suggestions.

Here is the answer to your requests.

Reviewer 2 Report

The paper is interesting because it concerns the study of an Aerostatic Pad Controlled by a Diaphragm-Valve. However, minor corrections are required.

  1. An Introduction should be improved. The Authors wrote,”…aerostatic bearings are particularly suitable for high precision applications.” This is fact, but some information should be added. It should be noted that aerostatic bearing is used in precise measuring instruments for example Talyrond 365, Taylor Hobson Company. This type of bearing does not generate additional vibrations, which significantly increases the measuring accuracy of this type of measuring systems. This is one of the many advantages of aerostatic bearings. Please refer to the following literature:

Ali, S.H.R. Method of optimal measurement strategy for ultra-high-precision machine in roundness nanometrology. Int. J. Smart Sens. Intell. Syst. 2015, 8, doi:10.21307/ijssis-2017-788.

Holub, M.; Jankovych, R.; Vetiska, J.; Sramek, J.; Blecha, P.; Smolik, J.; Heinrich, P. Experimental study of the volumetric error effect on the resulting working accuracy-Roundness. Appl. Sci. 2020, 10, doi:10.3390/APP10186233.

Stepien, K. In situ measurement of cylindricity - Problems and solutions. Precis. Eng. 2014, 38, doi:10.1016/j.precisioneng.2014.02.007.

Zmarzły, P. Multi-Dimensional Mathematical Wear Models of Vibration Generated by Rolling Ball Bearings Made of AISI 52100 Bearing Steel. Materials 2020, 13, 5440. https://doi.org/10.3390/ma13235440

  1. Page 11, Section 3

Please add more information about Sensitivity Analysis. Was this analysis simulation (computer-based)? If yes, please describe the software.

  1. Page 11, lines 237 and 239

Why these values of parameters were used in the analysis (supply pressure (?s=0.7 MPa), bearing size (?=40 mm), valve nozzle (?v=0.5 mm and ?p=1 mm).

  1. Page 17, Lines 281-283

The author wrote: “Also in this case, increasing the size of the pad integrated does not reduce the effectiveness of the method since this size variation is compensated by a variation of the diaphragm stiffness (Figure 18).

Please explain what this phenomenon results from?

  1. Page 20, Concussion

The conclusion should be improved. First, can the Authors indicate the optimal parameters (obtained from research) in relation to the specific application of aerostatic bearings? Such practical information would be useful to a wider audience.

Moreover, in the conclusion, there is a lacks of information on the direction of further research. Have the Authors considered conducting experimental tests to verify the results of simulation tests?

Author Response

Dear Reviewer,

Thank you for your valuable comments.

Here you can find the answers to your requests.
